# Cost-effectiveness of stem cell therapy versus standard of care for acute and subacute ischemic stroke

**Soichiro Takamiya**[1]☉, **Yasuhiro Morii**[2]☉, **Toshiya Osanai**[1]*, **Kazuki Ohashi**[3], **Katsuhiko Ogasawara**[3,4], **Kiyohiro Houkin**[5], **Miki Fujimura**[1]

**1** Department of Neurosurgery, Faculty of Medicine and Graduate School of Medicine, Hokkaido University, Sapporo, Japan, **2** National Institute of Public Health, Wako, Japan, **3** Faculty of Health Sciences, Hokkaido University, Sapporo, Japan, **4** Faculty of Engineering, Muroran Institute of Technology, Muroran, Japan, **5** Hokkaido University, Sapporo, Japan

☉ These authors are co-first authors and contributed equally to this work
* osanait@med.hokudai.ac.jp

## Abstract

A recent meta-analysis demonstrated the efficacy of stem cell therapy for ischemic stroke; however, its cost-effectiveness has not yet been sufficiently explored. Building on the clinical data from that meta-analysis, we aimed to evaluate the cost-effectiveness of the administration of mesenchymal stem cells for acute and subacute ischemic stroke compared with the standard of care. A cost-utility analysis was performed via simulation by using a Markov model. The participants were patients treated for acute or subacute ischemic stroke in hospitals in Japan. Stem cell therapy plus the standard of care (the stem cell group) was compared with the standard of care alone (the control group). The time horizon was 10 years. The primary outcome was the cost of stem cell therapy when the incremental cost-effectiveness ratio, calculated based on costs and quality-adjusted life years for both groups, was 5 million yen/quality-adjusted life year, the reference value in the cost-effectiveness evaluation in Japan. Efficacy data are obtained from the meta-analysis. Base-case analysis was performed using the most plausible or average values for all input parameters. Scenario and sensitivity analyses were also conducted. In the base case, the stem cell therapy cost when the incremental cost-effectiveness ratio was 5 million yen/quality-adjusted life year was $,3,746 from the public health payer's perspective and $5,157 from the public healthcare and long-term care payer's perspective. The scenario and sensitivity analyses supported the cost-effectiveness of stem cell therapy for acute and subacute ischemic stroke. This study provides threshold costs of stem cell therapy at which it becomes cost-effective for acute and subacute ischemic stroke. These results may support rational pricing strategies for stem cell therapy and facilitate its seamless integration into clinical practice.

**Data availability statement:** The data used in this study were obtained from our previously published meta-analysis. All data are publicly available at the following link: https://www.nature.com/articles/s41598-025-04405-6 (DOI: 10.1038/s41598-025-04405-6).

**Funding:** The author(s) received no specific funding for this work.

**Competing interests:** The authors have declared that no competing interests exist.

## Introduction

Ischemic stroke is a major cause of disability worldwide. In 2016, the estimated disability-adjusted life years due to ischemic stroke were 51.9 million [1]. Although its related mortality rate is lower than that related to hemorrhagic stroke, approximately 40% of patients with ischemic stroke experience moderate to severe sequelae that impair their independence [2]. Consequently, the cost of long-term care related to ischemic stroke may be higher than that related to other diseases. Additionally, great economic losses result when productive-age patients cannot return to their work [3].

Acute reperfusion therapies, such as intravenous thrombolysis and mechanical thrombectomy, have demonstrated numerous benefits for acute ischemic stroke. However, they are not always applicable or successful due to limited time windows or incomplete recanalization [4,5]. Stem cell therapy (SCT) is a promising option to promote neurological recovery in such cases [6]. Unlike reperfusion therapies that focus on restoring blood flow, the biological basis of SCT involves multifaceted mechanism, including neuroprotection, modulation of the inflammatory response, and stimulation of angiogenesis and endogenous neural repair [7–9]. A recent meta-analysis of randomized controlled trials (RCTs) demonstrated its efficacy for patients with acute and subacute ischemic stroke. Specifically, the proportion of patients who had attained a good neurological outcome (modified Rankin Scale [mRS] score = 0–2), by 90 days after treatment was higher in the SCT group (32.6%) than in the control group (26.1%) (risk ratio: 1.31; 95% confidence interval: 1.01–1.70) [10].

The economic impact of SCT has not been fully explored owing to its nascent stage. In one study, the sociological costs of SCT for ischemic stroke were estimated using hypothetical SCT effects based on expert opinion [11]. However, a cost-effectiveness analysis based on the actual efficacy of SCT for ischemic stroke has not yet been conducted. Additionally, in that study, the mRS score was assumed to remain unchanged from 3 months after the onset of stroke until death. However, that assumption might have been inappropriate because a longitudinal evidence suggests that functional recovery does not necessarily plateau at 3 months [12]. Additionally, long-term changes in mRS have been examined in other cost-effectiveness studies [13,14]. To address this limitation, simulation approaches that are more clinically relevant—such as the Markov model, which allows for transitions between health states—are warranted [15]. The clinical application of SCT for ischemic stroke is approaching, necessitating an analysis of its cost-effectiveness compared with that of the standard of care (SOC) for ischemic stroke by using data generated in clinical trials. This study therefore aimed to determine whether SCT is cost-effective compared with SOC from the perspective of the Japanese public healthcare payer. We hypothesized that SCT would be cost-effective if its price remains within a certain threshold, recognizing that cost-effectiveness depends on specific pricing and clinical outcome assumptions.

## Materials and methods

This study was conducted according to the Consolidated Health Economic Evaluation Reporting Standards (CHEERS) 2022 guidelines. The checklist is presented in S1

Table in S1 File. This study utilized only publicly available, previously published data and did not involve human subjects or any identifiable personal information. Accordingly, neither institutional review board approval nor informed consent was required.

## Participants and settings

The target population was patients treated in hospitals in Japan for acute or subacute stroke. SCT plus the SOC (the SCT group) was compared with SOC alone (the control group). A cost-utility analysis was performed using a Markov model from the public health payer's perspective. The public health payer's perspective, which included direct medical care costs based on the medical insurances in Japan is recommended in the guideline for cost-effectiveness evaluation in Japan. Universal fees for each medical service throughout Japan. The guideline also allows public healthcare and long-term care payer's perspective, which includes direct long-term care costs based on long-term care insurances as well as direct medical costs aside from the public health payer's perspective If the effect on public long-term care costs is important [16], we conducted another analysis from the perspective of the public healthcare and long-term care payer. For this analysis, we included both public medical and long-term care costs, as patients with severe ischemic stroke require nursing care for severe disabilities. The analyses were performed over a 10-year time horizon, representing the simulated duration of the model. The primary outcome was the SCT cost when the incremental cost-effectiveness ratio (ICER) was 5 million yen per quality-adjusted life year (QALY), as SCT for ischemic stroke is not yet reimbursed in Japan (i.e., the aforementioned medical cost in the SCT group did not include the SCT costs). The ICER was calculated using costs in the SCT and control groups ($Cost_{SCT}$ and $Cost_{control}$), the SCT costs, and QALYs in the SCT and control groups ($QALY_{SCT}$ and $QALY_{control}$). The reference value of 5 million yen/QALY has been used for official health technology assessments in Japan. The SCT cost was calculated by substituting the results obtained below into the following equation:

$$ICER = \frac{Increment\ of\ cost}{Increment\ of\ QALYs} = \frac{(Cost_{SCT} + SCT\ costs) - Cost_{control}}{QALY_{SCT} - QALY_{control}} = 5,000,000$$

The discount rate for the costs and outcomes was 2%/year in the base case. The costs were converted to US dollars by using the currency rate on February 6, 2026 ($1 = ¥157.2) [17].

## Markov model

A Markov model was used to estimate patients' status from months 4–120. Markov modeling is commonly used for chronic conditions, in which patients' health conditions change overtime [18,19]. Also, Markov modeling is commonly used for stroke [20,21], especially for post-acute stages. As mentioned earlier, the proportions of mRS scores until 3 months were obtained from our previous meta-analysis [10]. The model had three health statuses: functional independence (mRS scores of 0–2), disability (3–5), and death (6), and the Markov cycle was 1 month, as previously reported [15]. We assumed that patients' status could change every month (Fig 1) according to monthly transition probabilities (S1 Table in S1 File) [15]. The initial input parameters were the distributions of mRS scores, as explained in the following section (Table 1), and those distributions in each health status were regarded as constant.

## Efficacy dataset

In this study, we used the data of the initial probability (i.e., the mRS score at 90 days) obtained from our recent meta-analysis [10]. In the base-case analysis, we focused on the results of the main analysis from that work, which included studies with at least one patient who underwent SCT within 30 days of stroke onset (Table 1). Briefly, this dataset comprised 13 randomized controlled trials involving a total of 872 patients. These trials primarily evaluated the safety and efficacy of mesenchymal stem cell transplantation in patients with acute or subacute ischemic stroke, focusing on

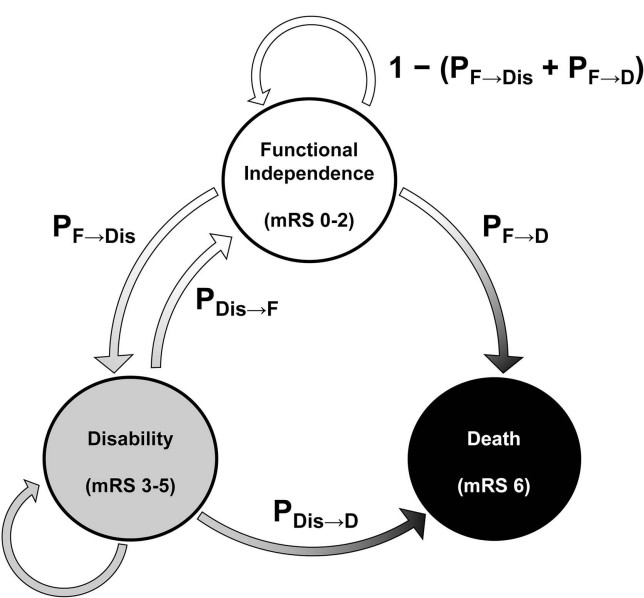

$$1 - (P_{F \to Dis} + P_{F \to D})$$

**Functional Independence (mRS 0-2)**

$$P_{F \to Dis} \qquad P_{Dis \to F} \qquad P_{F \to D}$$

**Disability (mRS 3-5)**

**Death (mRS 6)**

$$P_{Dis \to D}$$

$$1 - (P_{Dis \to F} + P_{Dis \to D})$$

**Fig 1. Schema of the Markov model.** The model had three health statuses: functional independence (mRS scores of 0–2), disability (3–5), and death (6). These statuses could change every month with monthly transition probabilities. mRS, modified Rankin scale; $P_{F \to Dis}$, probability of transition from functional independence to disability; $P_{F \to D}$, probability of transition from functional independence to death; $P_{Dis \to F}$, probability of transition from disability to functional independence; $P_{Dis \to D}$, probability of transition from disability to death.

**Table 1. Proportion of mRS stages at 3 months in the Markov model derived from the meta-analysis by Osanai et al. [10].**

|  | SCT group | Control group |
|---|---|---|
| Base case (all studies in the meta-analysis) |  |  |
| mRS score: 0–2 | 32.6% (105/322) | 26.1% (67/257) |
| mRS score: 3–5 | 63.9% | 70.4% |
| mRS score: 6 | 3.5% (16/457) | 3.5% (16/457) |
| Scenario analysis 1 (studies with SCT ≤ 30 days of stroke onset) |  |  |
| mRS score: 0–2 | 35.7% (101/283) | 26.9% (65/242) |
| mRS score: 3–5 | 60.3% | 69.1% |
| mRS score: 6 | 4.0% (16/403) | 4.0% (16/403) |
| Scenario analysis 2 (studies conducted in Japan) |  |  |
| mRS score: 0–2 | 34.1% (44/129) | 21.4% (24/112) |
| mRS score: 3–5 | 62.2% | 74.9% |
| mRS score: 6 | 3.7% (9/242) | 3.7% (9/242) |

The proportion of patients with mRS 3–5 (disability) was calculated as a residual by subtracting the proportions of mRS 0–2 (functional independence) and mRS 6 (death) from 100%; therefore, absolute patient numbers for mRS 3–5 are not directly available.

mRS, modified Rankin Scale; SCT, stem cell therapy

functional outcomes as measured by the mRS. The meta-analysis revealed a significant difference in the proportions of patients with functional independence by 90 days post-treatment between the SCT and control groups (RR = 1.31 [95% CI = 1.01–1.70]; p = 0.044; I2 = 0%).

For the scenario analyses of this study, we set two scenarios. For scenario 1, we used supplementary data from a sensitivity analysis presented in S2 Fig in S1 File of the supplementary material in the previous study [10], which included only studies in which all participants received SCT within 30 days of stroke onset. This scenario was set to ensure consistency in treatment timing, as the therapeutic effect of SCT may vary depending on when it is administered [22]. For scenario 2, we reanalyzed previous data obtained only from studies conducted in Japan, as the cost-effectiveness analysis in this study was based on the Japanese medical system. The methodology of this meta-analysis was the same as that in our previous study [10]. Further details regarding the methodology are provided in the Supplementary Material. In scenario 1, the proportions of patients with functional independence were similar to those for the main analysis—significantly different between the SCT and control groups [10]. Regarding scenario 2, the proportion of patients with functional independence by 90 days in the SCT group was also higher than that in the control group (34.1% vs. 21.4%, P = 0.04, S1 Fig in S1 File). Additionally, we performed a post-hoc analysis to reanalyze the proportions of patients who had died by 90 days, addressing the inconsistency of time points in previous mortality data. Specifically, we extracted trials that reported 90-day mortality from the 13 randomized controlled trials (RCTs) included in our original systematic review [10]. Pairwise meta-analyses of the extracted trials were performed as a post-hoc analysis of the proportions of patients with a modified Rankin Scale (mRS) score of 0–2 and 6 by 90 days between the SCT and control groups. The effect measure was the risk ratio (with 95% confidence intervals [CIs]). The I-squared statistic (I2) was used to assess heterogeneity among the included trials. A random-effects model was applied when I2 exceeded 25%; otherwise, a fixed-effects model was used. Additionally, sensitivity analyses were performed on studies in which all patients underwent SCT within 30 after the onset of stroke (scenario 1) or on studies that were conducted in Japan (scenario 2). Statistical analyses were performed using R version 4.3.2 (R Foundation for Statistical Computing, Vienna, Austria). Statistical significance was set at P < 0.05. Regarding scenario 2, among the 13 studies included in the main analysis, we extracted 6 [23–28]. Among these, two studies were performed in Japan [26,27]. In this specific scenario, the proportion of patients who attained an mRS score of 0–2 in the SCT group (44/129 patients, 34.1%) was significantly higher than that in the control group (24/112 patients, 21.4%; risk ratio, 1.60 [95% CI, 1.03–2.49], P = 0.04, S1 Figure in S1 File). The proportions of patients with an mRS score of 6 did not significantly differ between the SCT (11/260 patients, 4.2%) and control (5/197 patients, 2.5%) groups (risk ratio, 1.57 [95% CI, 0.63–3.92], P = 0.34; S2 Figure in S1 File). The results of the sensitivity analyses were similar; the proportions of patients with an mRS score of 6 also did not significantly differ between the SCT (11/221 patients, 5.0%) and control (5/182 patients, 2.7%) groups in scenario 1 (risk ratio, 1.70 [95% CI, 0.65–4.42], P = 0.28; S3 Figure in S1 File), or between the SCT (6/130 patients, 4.6%) and control (3/112 patients, 2.7%) groups in scenario 2 (risk ratio, 1.67 [95% CI, 0.48–5.81], P = 0.42; S4 Figure in S1 File).

As no difference was observed between the groups (main analysis: 4.2% vs. 2.5%, P = 0.34; scenario 1: 5.0% vs. 2.7%, P = 0.28; and scenario 2: 4.6% vs. 2.7%, P = 0.42; S2–S4 Figs in S1 File), we used a weighted average of 3.5% for the base case, 4.0% for scenario 1, and 3.7% for scenario 2 for both groups. Subsequently, the percentage of patients with disability was calculated as the remainder after the proportions of those with functional independence and those who had died were subtracted from 100%. The initial probabilities—which represent the proportions of mRS scores at 3 months, the starting point of the Markov model—are summarized in Table 1. Specifically, these probabilities show the likelihood of patients falling into different functional categories (based on their mRS scores) at the beginning of our analysis.

## Cost

The costs were estimated from published sources based on the patient disease severity obtained from the Markov model. In this study, medical costs other than those for stem cells themselves were assumed to be the same

between the SCT and control groups if patients had the same mRS scores. This assumption was based on the fact that all patients in the SCT group were administered stem cells intravenously or intra-arterially [23–29], requiring few additional costs. Medical costs during the hospitalization according to mRS scores at discharge were obtained from the Japan Stroke Data Bank 2015 [30] and adjusted with the consumer price index for 2015–2023 by multiplying by 104.7/97.8 (Table 2) [31]. As the Markov model in this study included only three statuses, the weighted average of medical costs for those statuses were calculated using the adjusted medical costs mentioned above and the distribution of mRS scores reported by Hattori et al., in which the ratio of mRS scores of 0:1:2 was 47:46:61, whereas that of mRS scores of 3:4:5 was 66:65:45 [32]. Consequently, the medical costs for mRS scores of 0–2, 3–5, and 6 were $8,597, $17,968, and $18,455, respectively. Only medical costs associated with the index hospitalization were included in this analysis. Since there has not been evidence that the long-term results such as rates of recurrent rates are different between the groups, Medical costs related to recurrence or long-term follow-up were considered the same and not included in the analysis.

**Table 2. Summary of parameters used in the analyses.**

| Parameters | | Base case | Range | Distribution | Sources |
|---|---|---|---|---|---|
| Efficacy and safety (for initial probability) | | | | | |
| Proportion of mRS scores of 0–2 for comparator (%) | | 26.1 | 10.0–36.1 | $\mu(0.261, 0.052)$ | Base [10] Upper [29] Lower [26] |
| Risk ratio of mRS scores of 0–2 | | 1.31 | 1.01–1.71 | $\mu(1.31, 0.15)$ | Base, Upper, Lower [10] |
| Proportion of mRS scores of 0–2 for comparator (%) | | 3.5 | 0.0–5.2 | $\mu(0.035, 0.007)$ | Upper [28] Lower [26] |
| Utility | | | | | |
| mRS scores of 0–2 | | 0.71 | 0.68–0.74 | $\beta(623.29, 254.58)$ | Base, Upper, Lower [33,34] |
| mRS scores of 3–5 | | 0.31 | 0.29–0.34 | $\beta(407.26, 906.49)$ | Base, Upper, Lower [33,35] |
| Costs | | | | | |
| Medical cost ($US/patient)[17] | mRS score of 0 | 5,866 | ±20% from base case | $\gamma(25, 2.715399e^{-05})$ | Base [28] Upper: base+20% Lower: base−20% |
| | mRS score of 1 | 7,899 | | $\gamma(25, 2.013142e^{-05})$ | |
| | mRS score of 2 | 11,236 | | $\gamma(25, 1.415299e^{-05})$ | |
| | mRS score of 3 | 12,939 | | $\gamma(25, 1.229076e^{-05})$ | |
| | mRS score of 4 | 15,528 | | $\gamma(25, 1.02423e^{-05})$ | |
| | mRS score of 5 | 21,9974 | | $\gamma(25, 7.229857e^{-06})$ | |
| | mRS score of 6 | 18,455 | | $\gamma(25, 8.617136e^{-06})$ | |
| Long-term care costs[a] ($US/patient)[21] | Support level 1 | 142 | ±20% from base case | $\gamma(25, 0.271688e^{-04})$ | Base [28] Upper: base+20% Lower: base−20% |
| | Support level 2 | 203 | | $\gamma(25, 1.249257e^{-04})$ | |
| | Care level 1 | 709 | | $\gamma(25, 2.179652e^{-05})$ | |
| | Care level 2 | 955 | | $\gamma(25, 1.469687e^{-05})$ | |
| | Care level 3 | 1,429 | | $\gamma(25, 9.40678e^{-06})$ | |
| | Care level 4 | 2,659 | | $\gamma(25, 7.998764e^{-06})$ | |
| | Care level 5 | 1,936 | | $\gamma(25, 7.116808e^{-06})$ | |
| Discount rate (%) | | 2 | 0–4 | | Base, Upper, Lower [16] |

[a]Other assumptions for estimation of long-term care costs are shown in S2 Table in S1 File

Long-term care costs were estimated based on patients' nursing support levels (1–2) and care levels (1–5) on a monthly basis from months 4–120, and the cumulative total was calculated. In Japan, when patients are certified as disabled, they are assigned a nursing support level according to their support/care level, which determines their eligibility for nursing care services under the nursing care insurance system. A higher nursing care level indicates greater severity of the patient's disability. This level was estimated based on the mRS score, as previously reported (S2 Table in S1 File) [13,15,36]. Individual long-term care costs for each nursing support/care level were calculated as the total annual care costs divided by the number of nursing care users in Japan, obtained from the survey of the Ministry of Health, Labour and Welfare (Table 2) [37]. The distributions of patients with care levels 2, 3, 4, and 5 were assumed to be in accordance with those in a previous study (i.e., care level 2:3 = 60.8:39.2, care level 4:5 = 61.0:39.0) [15]. The percentage of nursing care service users was also estimated according to the mRS score (S2 Table in S1 File) [13]. Long-term care costs were expected to be $0 for patients with mRS scores of 0 and 6, as these scores represent complete independence and death, respectively.

The medical and long-term care costs calculated above were used in the base-case analysis, which represents the most likely clinical scenario based on current clinical evidence as described above.

### Quality-adjusted life years

Utility scores were obtained according to the mRS scores based on previous reports [14,33]. Utility scores for each mRS score were as follows: for mRS scores of 0–2, 0.71; for scores of 3–5, 0.31; and for a score of 6, 0. These were used in the base-case analysis.

### Sensitivity analyses

Sensitivity and scenario analyses were also conducted to address the uncertainty of cost-effectiveness. The sensitivity analyses comprised one-way deterministic and probabilistic sensitivity analysis (DSA and PSA, respectively). The DSA varied one parameters at a time and PSA varied the related parameters at the same time to quantify the impact of parameter uncertainty on the results of a cost-utility analysis.The parameters and the sensitivity ranges for the DSA and the distributions of parameters for the PSA are shown in Table 2. The number of iterations for the PSA was 1000. In each iteration of PSA, values of the parameters were determined randomly based on the assumed distribution such as beta and gamma (Table 2).

The 5th and 95th percentiles of SCT costs at the reference value were provided as the uncertainty range.

## Results

### Cost

In the base case, the average medical cost without the SCT cost per patient was $14,504 in the SCT group and $15,069 in the control group. The average nursing care cost per patient over 10 years (months 4–120) was $43,242 in the SCT group and $44,652 in the control group. Consequently, the average total incremental cost per patient without the SCT cost was -$565 from the public health payer's perspective and -$1,976 from the public healthcare and long-term care payer's perspective. In scenario analyses 1 and 2, the average medical costs per patient were $14,239 and $14,375, respectively, in the SCT group, compared to $21,262 and $21,565, respectively, in the control group. The average nursing care costs for scenarios 1 and 2 were $42,308 and $42,812, respectively, in the SCT group, versus $44,217 and $45,568, respectively, in the control group. As a result, the average total incremental costs per patient for scenarios 1 and 2 were -$7,023 and -$7,190, respectively, from the public health payer's perspective, whereas they were -$8,932 and -$9,945, respectively from the public healthcare and long-term care payer's perspective (Table 3).

**Table 3. Costs, gained QALYs, and SCT costs in the base-case and scenario analyses.**

| Scenario | Medical cost ($US) | Long-term care cost ($US) | Total cost ($US) | Gained QALYs | SCT cost from public health payer's perspective ($US) when ICER is at the reference value | SCT cost from public healthcare and long-term care payer's perspective ($US) when ICER is at the reference value |
|---|---|---|---|---|---|---|
| Base case SCT Control Δ | | | | | | |
| | 14,504 | 43,242 | 57,746 | 2.17 | | |
| | 15,069 | 44,652 | 59,721 | 2.07 | | |
| | −565 | −1,410 | −1,976 | 0.10 | 3,746 | 5,157 |
| Scenario 1 SCT Control Δ | | | | | | |
| | 14,239 | 42,308 | 56,547 | 2.21 | | |
| | 21,262 | 44,217 | 65,480 | 2.07 | | |
| | −7,023 | −1,910 | −8,932 | 0.14 | 11,329 | 13,238 |
| Scenario 2 SCT Control Δ | | | | | | |
| | 14,375 | 42,812 | 57,187 | 2.19 | | |
| | 21,262 | 45,568 | 67,133 | 1.99 | | |
| | −7,0230 | −2,756 | −9,945 | 0.20 | 13,404 | 16,169 |

ICER, incremental cost-effectiveness ratio; QALY, quality-adjusted life year.

## QALYs

In the base case, the estimated average QALYs gained over 10 years were 2.17 in the SCT group and 2.07 in the control group, resulting in incremental QALYs of 0.10. In scenario analyses 1 and 2, the average gained QALYs were 2.21 and 2.19, respectively, in the SCT group, whereas they were 2.07 and 1.99, respectively, in the control group. Accordingly, the incremental QALYs in these scenarios were 0.14 and 0.20, respectively (Table 3).

## SCT cost at the reference ICER value

In the base case, the cost for SCT when the ICER was 5 million yen/QALY was $3,746 from the public health payer's perspective, and $5,157 from the public healthcare and long-term care payer's perspective. In scenario analyses 1 and 2, the SCT costs were $11,329 and $13,404, respectively, from the public health payer's perspective, and $13,328 and $16,160, respectively, from the public healthcare and long-term care payer's perspective (Table 3, Figs 2 and 3).

## Sensitivity analyses

The results of DSAs from the public health payer's perspective are summarized in S3 Table in S1 File and Fig 2. The SCT cost was most affected by the risk ratio of the proportion of patients with functional independence, indicating that the efficacy of SCT had the greatest impact on the SCT cost (range: $151–$10,531). The percentage of patients with functional independence in the control group also had a significant impact on SCT cost (range: $1,921–$10,234).

The results of DSAs from the public healthcare and long-term care payer's perspective are summarized in S4 Table in S1 File and Fig 3. The results were similar to those from the public health payer's perspective—the risk ratio of the proportion of patients with functional independence and the percentage of patients with functional independence in the control group significantly affected SCT cost (ranges: $207–$14,494 and $2,645–$13,791, respectively).

The 5th to 95th percentile range of the incremental QALYs was 0.03–0.25. The 5th to 95th percentile ranges of incremental costs without the SCT cost and the SCT cost when the ICER was the reference value were -$8704 to -$560 and $770–$9,040, respectively, from the public health payer's perspective, and -$4,359 to -$475 and $1,380–$11,988, respectively, from the public healthcare and long-term care payer's perspective.

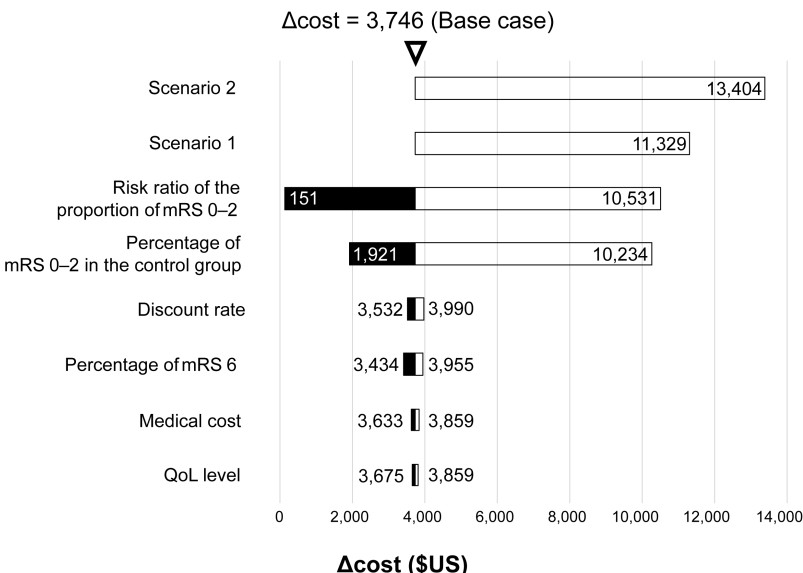

**Fig 2. Tornado chart for scenario and sensitivity analyses from the public health payer's perspective.** mRS, modified Rankin Scale; QoL, quality of life.

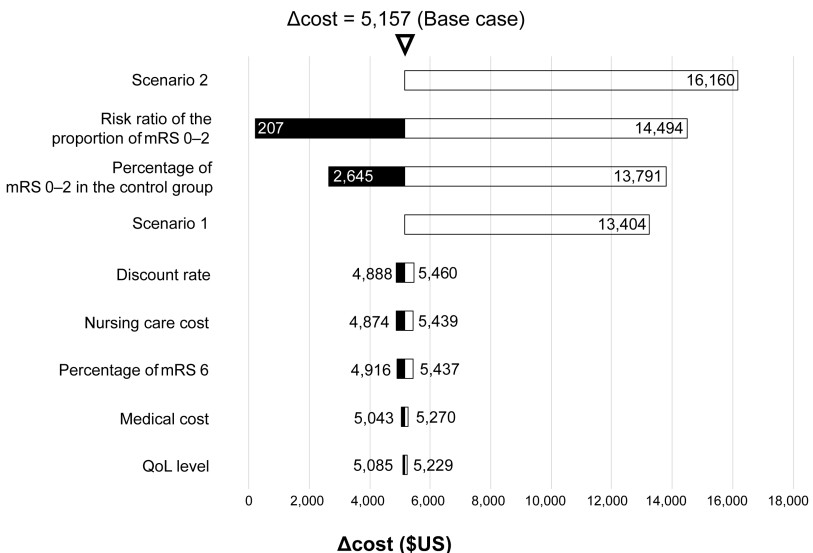

**Fig 3. Tornado chart for scenario and sensitivity analyses from the public healthcare and long-term care payer's perspective.** mRS, modified Rankin Scale; QoL, quality of life.

## Discussion

This was the first report of the cost-effectiveness of SCT for acute and subacute ischemic stroke based on the results of a meta-analysis [10] that provided the latest robust evidence regarding the efficacy of SCT for ischemic stroke. For the cost-effectiveness analysis, we estimated costs based on the medical system in Japan, which faces the challenge of a

super-aging society. The results demonstrate the range of SCT costs at which the cost-effectiveness of SCT for patients with acute and subacute ischemic stroke becomes superior to that of SOC treatment alone. Our sensitivity analyses also suggest that the SCT cost can be higher when stem cells are administered earlier (scenario 1) or used in a population with a higher proportion of patients with functional independence.

This is the first report to demonstrate the cost-effectiveness of SCT for ischemic stroke based on clinical data generated in RCTs. The RCTs included in the meta-analysis upon which this study is based involved the administration of stem cells, such as mesenchymal stem cells, via intravenous or intraarterial routes to patients in the acute or subacute phase of ischemic stroke. These trials collectively showed that SCT significantly increases the probability of achieving functional independence (modified Rankin Scale [mRS] scores of 0–2) by 90 days compared with SOC (32.6% vs. 26.1%; risk ratio: 1.31). Furthermore, the clinical evidence confirmed the safety of the intervention, with no significant increase in mortality rates observed in the SCT group across the trials (weighted average: 3.5% in both groups). By incorporating these robust data on both efficacy and safety from multiple RCTs, rather than relying on hypothetical assumptions, our analysis provides a clinically grounded evaluation of the economic impact of SCT [10]. Previously, only one cost-effectiveness study of SCT for ischemic stroke was reported. Svensson et al. reported that SCT reduced costs by $19,055 compared with the SOC [11]. However, that estimate was based on several assumptions, such as that SCT would improve the mRS score by 1 in 50% of patients and that the mRS score would not change from 3 months after stroke onset until death. Clinical trials have indicated that those assumptions are unrealistic [23–29,34,35,38–41]. We addressed this issue by incorporating results from a meta-analysis of several RCTs and applying a Markov model, considering changes in patients' health status. Therefore, our results are likely more realistic and clinically relevant.

As drug costs have been rapidly rising worldwide, cost-effectiveness analyses have become increasingly important [42,43]. This is especially crucial for novel treatments because it can serve as a basis for price-setting. Highly prevalent diseases, including ischemic stroke, can easily place a financial burden on governments if treatment costs are disproportionately high, as such treatments are expected to be widely used and may substantially increase healthcare expenses. Therefore, the cost-effectiveness analysis of treatments for common diseases should be especially rigorous [44]. Now that the efficacy of SCT for ischemic stroke is gradually being established, and clinical application may be just around the corner, we believe that this study provides valuable insights by evaluating it from an economic perspective [10]. While STEMIRAC® therapies for traumatic spinal cord injury often cost over $9,000 [45], our analysis suggests a significantly lower threshold of $3,746–$5,157 for SCT in ischemic stroke. This disparity reflects the high prevalence and substantial long-term care burden of stroke compared to diseases with no effective treatment available. Pricing a treatment for such a "common disease" at the level of drugs for diseases with no alternative treatment would be fiscally unsustainable for national healthcare systems. Consequently, a pricing strategy that balances clinical efficacy with the large volume of potential recipients is essential for the sustainable implementation of SCT in vascular neurology.

Certain limitations of this study must be noted. First, the meta-analysis on which this study is based revealed heterogeneity in terms of stem cell types, the timing of SCT, and dosage of administered cells, which might have resulted in an over- or underestimation of the efficacy of SCT. As more clinical trials on SCT for ischemic stroke are completed, evaluations will become more accurate. Second, this study was based on medical and long-term care costs in Japan. Consequently, while our findings are rooted in the Japanese healthcare system, their extrapolation to other regions—such as the UK, US, or major Asian economies—requires caution, as healthcare systems and economic thresholds differ significantly across countries. Furthermore, since no cell-based therapy for stroke has yet been clinically approved in these regions, real-world comparative data remain limited, making direct application to other socio-economic contexts challenging. However, as many countries are expected to experience aging populations in the near future [46], we believe that these results are important on a global scale.

## Conclusions

In the base-case scenario, if the price of SCT is < $3,746, it is superior to SOC for acute and subacute ischemic stroke from the public health payer's perspective. However, if the price of SCT is < $5,157, it still exhibits superior cost-effectiveness from the public healthcare and long-term care payer's perspective. We hope that this study will contribute to the seamless clinical application of SCT for patients with ischemic stroke. Notably, our scenario analyses demonstrated that the threshold cost for SCT increases significantly when clinical efficacy is maximized. These scenarios highlight that the economic value of SCT is highly sensitive to treatment timing and the specific clinical setting. By providing a range of threshold costs under different clinical conditions, this study offers a more robust and comprehensive framework for rational pricing strategies and facilitates the seamless integration of SCT into clinical practice in various healthcare environments.

## Supporting information

**S1 File. Supporting information including forest plots comparing the risks of modified Rankin Scale scores of 0–2 (S1 Fig) and mortality (S2–S4 Figs) at 90 days after treatment, monthly transition probabilities for the Markov model (S1 Table), nursing care levels (S2 Table), results of deterministic sensitivity analyses (S3–S4 Tables), CHEERS 2022 checklist, and output data from the Markov model for the placebo and SCT arms.**
(DOCX)

## Acknowledgments

We would like to acknowledge Editage's (www.editage.com) support in manuscript preparation.

## Author contributions

**Conceptualization:** Toshiya Osanai, Miki Fujimura.

**Data curation:** Soichiro Takamiya, Yasuhiro Morii.

**Methodology:** Toshiya Osanai, Kazuki Ohashi, Katsuhiko Ogasawara.

**Supervision:** Kiyohiro Houkin.

**Writing – original draft:** Soichiro Takamiya.

**Writing – review & editing:** Yasuhiro Morii.

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
