## [Decision Letter · Decision Letter 0]

15 Dec 2025

PONE-D-25-56009Cost-Effectiveness of Stem Cell Therapy versus Standard of Care for Acute and Subacute Ischemic StrokePLOS One

Dear Dr. Osanai,

Thank you for submitting your manuscript to PLOS ONE. After careful consideration, we feel that it has merit but does not fully meet PLOS ONE’s publication criteria as it currently stands. Therefore, we invite you to submit a revised version of the manuscript that addresses the points raised during the review process.

We look forward to receiving your revised manuscript.

Kind regards,

Nazmul Haque

Academic Editor

PLOS One

Journal Requirements:

When submitting your revision, we need you to address these additional requirements

Reviewers' comments:

Reviewer's Responses to Questions

**Comments to the Author**

1. Is the manuscript technically sound, and do the data support the conclusions?

Reviewer #1: Yes

Reviewer #2: Yes

2. Has the statistical analysis been performed appropriately and rigorously? 

Reviewer #1: Yes

Reviewer #2: I Don't Know

3. Have the authors made all data underlying the findings in their manuscript fully available?

Reviewer #1: Yes

Reviewer #2: Yes

4. Is the manuscript presented in an intelligible fashion and written in standard English?

Reviewer #1: Yes

Reviewer #2: Yes

5. Review Comments to the Author

Reviewer #1: Takamiya and Morii et al. demonstrated the range of SCT costs at which the cost- effectiveness of SCT for patients with acute and subacute ischemic stroke becomes superior to that of SOC treatment alone. I believe this is a valuable study in terms of clarifying the importance of stem cell treatment in a statistical manner. Moreover, I agree that choosing the medical system in Japan to estimate costs for the cost-effectiveness analysis is a good idea since Japanese health care system is being challenged with aging population. However, my opinion is that this study should be easily read by patients, their relatives, and the law makers (since it has crucial analyses for stem cell treatment) in addition to our scientific community. In this manner, I highly recommend the authors to explain biostatistical terms in more detail in this paper where it is possible to make this paper more easily understandable.

I am hoping that my recommendations below will improve this paper.

-Table 1. 1) Please clearly explain in the table title that what are these percentages.

2) Please add patient numbers for mRS score 3-5 in the base, scenario 1 and 2.

3) Please add short explanations on what the base, scenarios 1 and 2 are on table 1 so that everybody can see what they stand for on the table quickly.

-Page 6 Line 130-135: too many uses of “that study” in writing. Please rewrite this paragraph so that it could be clear for everybody.

-Page 7 Line 156. Do not assume that only biostatisticians are going to read your paper and please explain what you mean with “Initial Probabilities in Table 1” for the people with little or no statistics knowledge. Your paper should be readable for general public, as well, since the topic of your paper is very important to the patients, their relatives, and policy makers to make a decision in these treatments.

-Table 2 is very unorganized in this current format. Please make sure that your table has a decent look. You may add fill all borders with lines so that we can see the columns and rows of the table clearly. Again, please explain what beta, gama, and care levels are in the title of the table.

-Page 10 Line 179: How did you decide nursing support and care levels? If there is a published standard, please cite. If not, please explain.

-Similar to Table 2, Table 3 is also unorganized. Please read my comment for Table 2 above and make rows and columns easily distinguishable.

-Figures 2 and 3: I recommend adding the percentage and the risk ratio of mRS 3-5 in figures 2 and 3 in the sensitivity analysis for the sake of being more informative in these graphs since there is a risk that the disease status of the patients might change from 0-2 to 3-5.

Discussion:

Page 16 line 288: Discuss what clinical trial demonstrated in more detail.

Conclusions:

Your conclusion is missing your scenarios. Please add them in your conclusion as well and tell us what the importance of those scenarios is in addition to mentioning threshold costs only for base-case scenario. Have a more comprehensive and inclusive conclusion.

-Author contributions are missing. Please clearly state who did what job in this paper.

Reviewer #2: Takamiya et al. have performed a cost-effectiveness analysis of stem cell therapy when compared to standard of care in the treatment of ischemic stroke. They have presented their findings in the context of the Japanese healthcare system and provide a set of threshold costs that would make stem cell therapy cost-effective for application in the treatment of acute/sub-acute ischemic stroke. While the study is interesting as it focuses on the financial implications of stem cell therapy in a condition that may see increased application in the near future, certain aspects of the manuscript require further clarification to make it easier to interpret for the readers.

Comments:

1. While the abstract is overall well-written, it is unclear when the authors refer to the meta-analysis that has been the basis of this manuscript. The authors should better present/explain this in the abstract.

2. In the introduction, there is sufficient focus on the economic perspective of stroke treatment. It will also be nice to include the biological basis of stem cell therapy for ischemic stroke, to put it into perspective for the readers why this comparison is being made with the standard of care. The standard of care should also be briefly described along with this (acute reperfusion therapies, line 52).

3. In line 58, please include the statistics as done in line 59.

4. In line 66, please elaborate on the statement that the “assumption is unrealistic”. Please cite suitable references in support of your statement(s).

5. At the end of the introduction, please clearly highlight the hypothesis and research questions being addressed by this study.

6. In line 79, there is an extra period symbol.

7. In line 86 and 91, the authors introduce the terminology of “public health payer’s perspective” and “public healthcare and long-term care payer”, respectively. Please elaborate on these terms so that it is also clear for a reader to interpret this who do not have a background in this field.

8. In line 107, the authors state that they use the currency rate as per December 2024. It should be relatively easy to adjust the conversion rates to a more recent period in 2025, to make the comparison more relevant.

9. In line 132, the authors state the dataset used in the main analysis of another study. Please describe this dataset in brief, to put it in context for the readers of the current manuscript.

10. In line 134, the authors state that there was a significant difference in the comparison. Please add the statistics to support this.

11. In line 138 and 144, please link the supplementary material accordingly as it is unclear that this has been highlighted there.

12. Please add references for the statement made in line 141 and briefly elaborate.

13. I would recommend that the authors move the methods and results text from the supplementary materials to the main manuscript. This will increase clarity of the manuscript. Please only include figures and tables in the supplementary material.

14. Please briefly elaborate on the statement in line 173 and 174 with suitable references.

15. Please provide suitable references for the statements in lines 181 – 184.

16. In line 195, please explain in brief about the base-case analysis used in this manuscript.

17. Please expand the acronym QALYs in the methods section header (line 198).

18. In the results section, I would request the authors to provide a “take home message” and the end of every result subsection that briefly describes the findings that they report. This will increase readability of the results.

19. In figures 2 and 3, also state the reported values for each variable in the tornado plot. This will increase the readability of the plots.

20. In the discussion, I would urge the authors to put into perspective the reported cost-effectiveness analysis for stem cell therapy in ischemic stroke with other cell based therapies. This will make it easier for the reader the interpret the reported financial aspects better.

21. I would also ask the authors to have a discussion of their findings from Japan in the context of other major healthcare systems from the UK-EU, US-Canada and major Asian economies (such as India, Singapore, China). Can these findings be extrapolated to other regions? Would any suitable adjustments be required to be made based on socio-economic features?

22. The authors should also discuss their choice of using a Markov model for the analysis in this manuscript. This may also be done in the discussion section. Please also put into context if similar statistical approaches have been used in prior studies that evaluate cost-effectiveness of different treatment groups. This may be done in either in the methods or discussion section.

23. Can the authors please add the output data from running the Markov models in the supplementary material? This will make it easier for the readers to follow and interpret the reported results.

6. PLOS authors have the option to publish the peer review history of their article (what does this mean?). If published, this will include your full peer review and any attached files.

Reviewer #1: No

Reviewer #2: No

---

## [Author Response · Author response to Decision Letter 1]

26 Feb 2026

０Journal Requirements:

When submitting your revision, we need you to address these additional requirements

Response： We sincerely appreciate your helpful guidance. The manuscript and all associated files have been thoroughly revised to ensure full compliance with PLOS ONE’s formatting and file naming requirements, following the provided templates.

Response：Thank you for your reminder and for providing the detailed guidelines on code sharing. Our R codes for the cost-utility analysis will be available upon reasonable requests.

Response：The reviewers did not mention any specific published works.

Reviewers' comments:

Reviewer's Responses to Questions

Comments to the Author

1. Is the manuscript technically sound, and do the data support the conclusions?

Reviewer #1: Yes

Reviewer #2: Yes

2. Has the statistical analysis been performed appropriately and rigorously?

Reviewer #1: Yes

Reviewer #2: I Don't Know

3. Have the authors made all data underlying the findings in their manuscript fully available?

Reviewer #1: Yes

Reviewer #2: Yes

4. Is the manuscript presented in an intelligible fashion and written in standard English?

Reviewer #1: Yes

Reviewer #2: Yes

5. Review Comments to the Author

Reviewer #1: Takamiya and Morii et al. demonstrated the range of SCT costs at which the cost- effectiveness of SCT for patients with acute and subacute ischemic stroke becomes superior to that of SOC treatment alone. I believe this is a valuable study in terms of clarifying the importance of stem cell treatment in a statistical manner. Moreover, I agree that choosing the medical system in Japan to estimate costs for the cost-effectiveness analysis is a good idea since Japanese health care system is being challenged with aging population. However, my opinion is that this study should be easily read by patients, their relatives, and the law makers (since it has crucial analyses for stem cell treatment) in addition to our scientific community. In this manner, I highly recommend the authors to explain biostatistical terms in more detail in this paper where it is possible to make this paper more easily understandable.

I am hoping that my recommendations below will improve this paper.

-Table 1. 1) Please clearly explain in the table title that what are these percentages.

Response：Thank you. We have revised the table title as “ Proportion of mRS stages at 3 months in the Markov model derived from the meta-analysis by Osanai et al.[10]”

2) Please add patient numbers for mRS score 3-5 in the base, scenario 1 and 2.

Response：Thank you for pointing this out. In this study, the proportion of patients with mRS 3–5 (patients with disability) was calculated as a residual value by subtracting the proportions of patients with functional independence (mRS 0–2) and death (mRS 6) from 100%. Therefore, the absolute patient numbers for mRS 3–5 cannot be directly derived from the available data. We agree that the inclusion of patient numbers for mRS 0–2 and 6 in Table 1 may be confusing, and we are willing to remove these numbers from Table 1 if deemed appropriate by the editor and reviewers.

3) Please add short explanations on what the base, scenarios 1 and 2 are on table 1 so that everybody can see what they stand for on the table quickly.

Response：Thank you very much for your suggestion. We have added following explanations to Table 1 to clarify the definitions of the base case and each scenario, so that readers can easily understand their meaning; Base case (all studies in the meta-analysis), Scenario analysis 1 (studies with SCT ≤30 days of stroke onset), and Scenario analysis 2 (studies conducted in Japan).

-Page 6 Line 130-135: too many uses of “that study” in writing. Please rewrite this paragraph so that it could be clear for everybody.

Response：Thank you for your valuable suggestion. We agree that the repetitive use of 'that study' was potentially confusing. We have revised the paragraph to clarify the distinction between the current study and the referenced meta-analysis. Specifically, we replaced 'that study' with 'the meta-analysis' and 'that work' to improve readability and precision.

-Page 7 Line 156. Do not assume that only biostatisticians are going to read your paper and please explain what you mean with “Initial Probabilities in Table 1” for the people with little or no statistics knowledge. Your paper should be readable for general public, as well, since the topic of your paper is very important to the patients, their relatives, and policy makers to make a decision in these treatments.

Response：We sincerely appreciate this comment. We recognize the importance of making our findings accessible to a broader audience, including patients and policy makers. Accordingly, we have revised the text to define 'initial probabilities' as the starting health status of patients following treatment. We have also added a brief explanation of how these probabilities represent the distribution of functional outcomes, ensuring the baseline of our model is clear to non-statisticians.

-Table 2 is very unorganized in this current format. Please make sure that your table has a decent look. You may add fill all borders with lines so that we can see the columns and rows of the table clearly.

Response：Thank you for your feedback. To enhance the visibility of Table 2, we have added borders to all cells. In the revised table, each row and column can now be clearly distinguished.

Again, please explain what beta, gama, and care levels are in the title of the table.

Response：Thank you for your comment. We have added description on what the beta and gamma mean and additional explanation on the sensitivity analysis in the “Sensitivity analyses” section.

-Page 10 Line 179: How did you decide nursing support and care levels? If there is a published standard, please cite. If not, please explain

Response：We had described that in our previous manuscript (Line 181-188 in our previous manuscript). The correspondence of mRS scores to nursing care levels were on Table S2. This is in accordance with previous studies. We have added reference information on the manuscript.

-Similar to Table 2, Table 3 is also unorganized. Please read my comment for Table 2 above and make rows and columns easily distinguishable.

Response：Thank you very much for your feedback. In response to your comments on Table 2, we have similarly revised the format of Table 3. In addition, we have applied the same adjustments to Table 1, including the addition of borders and layout reorganization. This ensures consistent readability across all tables.

-Figures 2 and 3: I recommend adding the percentage and the risk ratio of mRS 3-5 in figures 2 and 3 in the sensitivity analysis for the sake of being more informative in these graphs since there is a risk that the disease status of the patients might change from 0-2 to 3-5.

Response：Thank you for your suggestion. We think the sensitivity analysis should be without the parameter mRS3-5. In the upper side of Figure 3, the impact of changes in parameters such as percentage of mRS0–2 in the control group. In this study, the proportion of mRS3-5 is defined as the complement of mRS0-2 and mRS6 (equal percentage between groups). Therefore, the effect of mRS0-2 and mRS3-5 are fundamentally the same because if the percentage of mRS0-2 increases by 1%, that of mRS3-5 decreases by 1%, and vice versa. Also, our meta-analysis show that the proportion of mRS0-2, not mRS3-5, increases by stem cell therapy. Therefore, it is appropriate that the impact of the proportion of and the risk ratio of mRS0-2 in the sensitivity analysis.

Discussion:

Page 16 line 288: Discuss what clinical trial demonstrated in more detail.

Response: Thank you for this insightful suggestion. We have revised the Discussion section to provide more specific clinical details from the RCTs used in our meta-analysis. Specifically, we have added information regarding the administration routes, the specific patient population (acute/subacute), and the key efficacy and safety results—such as the 32.6% vs. 26.1% achievement rate of mRS 0–2 and the 3.5% mortality rate. We believe these additions clarify the clinical foundation of our cost-effectiveness model ( Lines 335- 346 in the revised manuscript).

Conclusions:

Your conclusion is missing your scenarios. Please add them in your conclusion as well and tell us what the importance of those scenarios is in addition to mentioning threshold costs only for base-case scenario. Have a more comprehensive and inclusive conclusion.

Response: We appreciate the reviewer’s valuable suggestion. We have significantly revised the Conclusion section to include the results in our scenario analyses. We have also added a discussion on the importance of these scenarios, emphasizing how treatment timing and regional data alignment influence the economic value of SCT. This provides a more comprehensive perspective on the therapy's cost-effectiveness.

-Author contributions are missing. Please clearly state who did what job in this paper.

Response：Thank you for the comment. While the PLOS ONE format typically does not require author contributions in a manuscript, we have added an "Author Contributions" section at the end of the manuscript in response to your suggestion. We leave the final decision regarding its inclusion in the final text to the Editor's discretion.

Reviewer #2: Takamiya et al. have performed a cost-effectiveness analysis of stem cell therapy when compared to standard of care in the treatment of ischemic stroke. They have presented their findings in the context of the Japanese healthcare system and provide a set of threshold costs that would make stem cell therapy cost-effective for application in the treatment of acute/sub-acute ischemic stroke. While the study is interesting as it focuses on the financial implications of stem cell therapy in a condition that may see increased application in the near future, certain aspects of the manuscript require further clarification to make it easier to interpret for the readers.

Comments:

1. While the abstract is overall well-written, it is unclear when the authors refer to the meta-analysis that has been the basis of this manuscript. The authors should better present/explain this in the abstract.

Response: Thank you for your insightful comment. We have revised the abstract to clarify the relationship between our study and the meta-analysis mentioned.

2. In the introduction, there is sufficient focus on the economic perspective of stroke treatment. It will also be nice to include the biological basis of stem cell therapy for ischemic stroke, to put it into perspective for the readers why this comparison is being made with the standard of care. The standard of care should also be briefly described along with this (acute reperfusion therapies, line 52).

Response: Thank you for your constructive suggestion. Following your advice, we have revised the Introduction to include the biological basis of stem cell therapy, highlighting its mechanisms such as neuroprotection and immunomodulation. We have also provided a brief description of the current standard of care, specifically mentioning acute reperfusion therapies (intravenous thrombolysis and mechanical thrombectomy) to clarify the clinical context and the rationale for our comparison.

3. In line 58, please include the statistics as done in line 59.

Response: Thank you for your comment. We apologize for the lack of clarity in the previous version of the manuscript. Given that this analysis is based on the risk ratio between the SCT and control groups, we have revised the text to present the frequency data for both the SCT group (32.6%) and the control group (26.1%) alongside the comparative statistics to facilitate a better understanding for the reader.

4. In line 66, please elaborate on the statement that the “assumption is unrealistic”. Please cite suitable references in support of your statement(s).

Response: Thank you for your comment. We have revised the Introduction to clarify why a functional plateau at 3 months is considered unrealistic and have added a supporting reference.

A longitudinal study by Lee et al. (2015) demonstrated that neurological and functional impairments—including ADL and gait—continue to improve significantly for up to 6 months post-stroke. Their multi-time-point analysis explicitly showed that recovery variables had not yet reached a plateau between 3 and 6 months. Additionally, other cost-effectiveness studies (references 24, 27) have examined long-term changes in mRS. Therefore, we used a Markov model to more accurately reflect these ongoing clinical transitions compared to a static model. We have also revised the term "unrealistic" to "inappropriate" to moderate the strength of the expression.

5. At the end of the introduction, please clearly highlight the hypothesis and research questions being addressed by this study.

Response: Thank you for your suggestion. We have revised the final paragraph of the Introduction to explicitly state our research question and hypothesis. Specifically, we added that this study aims to investigate the cost-effectiveness of SCT from the Japanese public payer's perspective, testing the hypothesis that SCT is cost-effective within certain price thresholds. We believe these additions clarify the objectives of our study while maintaining the original context.

6. In line 79, there is an extra period symbol.

Response：Thank you for pointing this out. We have corrected the typographical error by removing the extra period in line 79 in the previous manuscript.

7. In line 86 and 91, the authors introduce

---

## [Decision Letter · Decision Letter 1]

23 Mar 2026

PONE-D-25-56009R1Cost-Effectiveness of Stem Cell Therapy versus Standard of Care for Acute and Subacute Ischemic StrokePLOS One

Dear Dr. Osanai,

Thank you for submitting your manuscript to PLOS ONE. After careful consideration, we feel that it has merit but does not fully meet PLOS ONE’s publication criteria as it currently stands. Therefore, we invite you to submit a revised version of the manuscript that addresses the points raised during the review process.

We look forward to receiving your revised manuscript.

Kind regards,

Nazmul Haque

Academic Editor

PLOS One

Journal Requirements:

Reviewers' comments:

Reviewer's Responses to Questions

**Comments to the Author**

1. If the authors have adequately addressed your comments raised in a previous round of review and you feel that this manuscript is now acceptable for publication, you may indicate that here to bypass the “Comments to the Author” section, enter your conflict of interest statement in the “Confidential to Editor” section, and submit your "Accept" recommendation.

Reviewer #1: All comments have been addressed

Reviewer #2: All comments have been addressed

2. Is the manuscript technically sound, and do the data support the conclusions?

Reviewer #1: Yes

Reviewer #2: Yes

3. Has the statistical analysis been performed appropriately and rigorously? 

Reviewer #1: Yes

Reviewer #2: Yes

4. Have the authors made all data underlying the findings in their manuscript fully available?

Reviewer #1: Yes

Reviewer #2: Yes

5. Is the manuscript presented in an intelligible fashion and written in standard English?

Reviewer #1: Yes

Reviewer #2: Yes

6. Review Comments to the Author

Reviewer #1: Thanks so much for addressing my points. I request that you address my last point below:

Table 1. Please explain under the table (along with the abbreviations) why you could not include direct patient numbers for mRS score 3-5 rather than deleting patient numbers for other mRS groups.

Reviewer #2: I thank the authors for revising the manuscript on the basis of my review. I find the manuscript to be acceptable for publication in its current form.

7. PLOS authors have the option to publish the peer review history of their article (what does this mean?). If published, this will include your full peer review and any attached files.

Reviewer #1: No

Reviewer #2: No

---

## [Author Response · Author response to Decision Letter 2]

23 Apr 2026

０Journal Requirements:

Response: No specific publications were recommended for citation by the reviewers. Accordingly, no additional references have been added on this basis.

Response: We have reviewed our reference list and confirmed that none of the cited works have been retracted. No changes to the reference list were necessary.

Reviewers' comments:

Reviewer's Responses to Questions

Comments to the Author

1. If the authors have adequately addressed your comments raised in a previous round of review and you feel that this manuscript is now acceptable for publication, you may indicate that here to bypass the “Comments to the Author” section, enter your conflict of interest statement in the “Confidential to Editor” section, and submit your "Accept" recommendation.

Reviewer #1: All comments have been addressed

Reviewer #2: All comments have been addressed

2. Is the manuscript technically sound, and do the data support the conclusions?

Reviewer #1: Yes

Reviewer #2: Yes

3. Has the statistical analysis been performed appropriately and rigorously?

Reviewer #1: Yes

Reviewer #2: Yes

4. Have the authors made all data underlying the findings in their manuscript fully available?

Reviewer #1: Yes

Reviewer #2: Yes

5. Is the manuscript presented in an intelligible fashion and written in standard English?

Reviewer #1: Yes

Reviewer #2: Yes

6. Review Comments to the Author

Reviewer #1: Thanks so much for addressing my points. I request that you address my last point below:

Table 1. Please explain under the table (along with the abbreviations) why you could not include direct patient numbers for mRS score 3-5 rather than deleting patient numbers for other mRS groups.

Response: Thank you for this suggestion. As recommended, we have added the following footnote to Table 1 to clarify why absolute patient numbers for mRS 3–5 could not be provided:

"The proportion of patients with mRS 3–5 (disability) was calculated as a residual by subtracting the proportions of mRS 0–2 (functional independence) and mRS 6 (death) from 100%; therefore, absolute patient numbers for mRS 3–5 are not directly available."

Reviewer #2: I thank the authors for revising the manuscript on the basis of my review. I find the manuscript to be acceptable for publication in its current form.

Response: We sincerely thank the reviewer for the thoughtful evaluation and constructive comments, which have helped improve the quality of our manuscript.

7. PLOS authors have the option to publish the peer review history of their article (what does this mean?). If published, this will include your full peer review and any attached files.

Do you want your identity to be public for this peer review? For information about this choice, including consent withdrawal, please see our Privacy Policy.

Reviewer #1: No

Reviewer #2: No

---

## [Decision Letter · Decision Letter 2]

5 May 2026

Cost-Effectiveness of Stem Cell Therapy versus Standard of Care for Acute and Subacute Ischemic Stroke

PONE-D-25-56009R2

Dear Dr. Osanai,

We’re pleased to inform you that your manuscript has been judged scientifically suitable for publication and will be formally accepted for publication once it meets all outstanding technical requirements.

Kind regards,

Nazmul Haque

Academic Editor

PLOS One

Additional Editor Comments (optional):

Reviewers' comments:

Reviewer's Responses to Questions

**Comments to the Author**

1. If the authors have adequately addressed your comments raised in a previous round of review and you feel that this manuscript is now acceptable for publication, you may indicate that here to bypass the “Comments to the Author” section, enter your conflict of interest statement in the “Confidential to Editor” section, and submit your "Accept" recommendation.

Reviewer #1: (No Response)

2. Is the manuscript technically sound, and do the data support the conclusions?

Reviewer #1: (No Response)

3. Has the statistical analysis been performed appropriately and rigorously? 

Reviewer #1: (No Response)

4. Have the authors made all data underlying the findings in their manuscript fully available?

Reviewer #1: (No Response)

5. Is the manuscript presented in an intelligible fashion and written in standard English?

Reviewer #1: (No Response)

6. Review Comments to the Author

Reviewer #1: (No Response)

7. PLOS authors have the option to publish the peer review history of their article (what does this mean?). If published, this will include your full peer review and any attached files.

Reviewer #1: No

---

## [Editor Report · Acceptance letter]

PONE-D-25-56009R2

PLOS One

Dear Dr. Osanai,

I'm pleased to inform you that your manuscript has been deemed suitable for publication in PLOS One. Congratulations! Your manuscript is now being handed over to our production team.

Kind regards,

on behalf of

Dr. Nazmul Haque

Academic Editor

PLOS One